# Silica Accumulation in Potato (*Solanum tuberosum* L.) Plants and Implications for Potato Yield Performance—Results from Field Experiments in Northeast Germany

**DOI:** 10.3390/biology13100828

**Published:** 2024-10-16

**Authors:** Daniel Puppe, Jacqueline Busse, Mathias Stein, Danuta Kaczorek, Christian Buhtz, Jörg Schaller

**Affiliations:** 1Leibniz Centre for Agricultural Landscape Research (ZALF), 15374 Müncheberg, Germany; 2Soil Science and Soil Protection, Martin Luther University Halle-Wittenberg, 06120 Halle (Saale), Germany; 3Department of Soil Science, Warsaw University of Life Sciences (SGGW), 02-776 Warsaw, Poland

**Keywords:** phytogenic silica, crop production, phytoliths, sustainability, biogenic silica, stress resilience, plant-available silicon, silica amendment, long-term field experiment

## Abstract

The potato is the most important non-cereal food crop worldwide. Silicon (Si) fertilizers have been reported to improve potato growth and yield. We used results from two field experiments in the temperate zone to gain insight into silica accumulation in potato plants as well as corresponding long-term potato yield performance. We found relatively low Si contents in potato plants grown in soils with different concentrations of plant-available Si (field experiment 1). Moreover, potato yield was not correlated to plant-available Si concentrations in soils in the long term (1965–2015, field experiment 2). Based on our results, we ascribe the reported positive effects of Si fertilization on potatoes rather to effects of the used Si fertilizers than to silica accumulation in potato plants. While Si fertilizers applied directly to the leaves can prevent fungal infections, soil-applied Si fertilizers can enhance phosphorus and water availability in agricultural soils. With our study, we aim to inspire further research on Si fertilization–potato relationships. The corresponding results will help to derive practice-oriented recommendations for potato growers worldwide to cope with the challenges of climate change.

## 1. Introduction

As silicon (Si) is the second most abundant element in the Earth’s crust, Si can be found virtually everywhere. Thus, it is not surprising that Si is also an important component in many organisms like protists, sponges, and plants, which use dissolved monomeric silicic acid (Si(OH)_4_) for the formation of biogenic silica (i.e., amorphous hydrated silica, SiO_2_·*n*H_2_O). This process of biosilicification has been found to represent a key factor in the global Si cycle [1,2,3]. In terrestrial ecosystems, Si cycling by vegetation has been the focus of research [4,5,6], although the role of protists (i.e., testate amoebae) has been highlighted since the beginning of the 21st century [7].

Precipitated biogenic silica in plants is called phytogenic silica, which can be found within cells (i.e., in the cell wall and the cell lumen) and in intercellular spaces and extracellular (cuticular) layers. While intercellular and extracellular phytogenic silica structures are quite delicate/fragile, cell wall and lumen silica precipitates are quite resilient and can persist in soils as microfossils (phytoliths) up to hundreds and thousands of years [8,9]. These phytoliths are routinely used in many scientific fields like archaeology, (paleo)botany, (evolutionary) biology, plant taxonomy, or climatology, and thus a phytolith nomenclature and classification system has evolved [10]. As phytoliths can also contain various elements like carbon, aluminum, calcium, iron, manganese, phosphorus, lead, copper, cadmium, or arsenic, their potential for carbon and metal(loid) long-term sequestration in soils has been recognized recently [11,12,13].

In general, Si accumulation in plants has been shown to enhance plants’ resistance to abiotic and biotic stress with implications for plant performance and ecosystem functioning [14]. In agricultural plant–soil systems, Si fertilizers are widely used to increase yields of Si-accumulating crops like rice, maize, wheat, and sugarcane, especially in the (sub)tropics, where soils usually are desilicated much stronger than in the temperate zone [15,16,17]. In this context, Si-rich materials used for fertilization comprise industrial waste matter (i.e., slags or silica fume), manufactured fertilizers (e.g., fused magnesium phosphate or potassium silicate), and minerals mined from the earth’s surface (e.g., wollastonite or diatomaceous earth) [15,18]. However, the production of these fertilizers is quite energy-consuming, and fertilization with some of these products can cause environmental problems (e.g., metal(loid) contamination of soils). Biochar has been discussed as a comparatively environmentally friendly Si source in agriculture [19], but it has to be considered that its production by pyrolysis of crop residues and manures is relatively CO_2_-intensive [20]. In the long term, the maximum restoration of the Si cycle in agricultural plant–soil systems by crop straw recycling might represent the most promising and environmentally friendly approach for the sustainable agricultural production of resilient crops [21,22].

The potato (*Solanum tuberosum* L., family Solanaceae) represents one of the most important crops worldwide. According to the Food and Agriculture Organization (FAO) of the United Nations, approximately 375 million tons of potatoes were produced worldwide in the year 2022 [23]. Despite the fact that plant species from the family Solanaceae are considered to be non-Si-accumulating [24], some studies reported beneficial effects of Si fertilization on potato production. Crusciol et al. [25], for example, found that Si application in a greenhouse pot experiment significantly increased potato tuber yield and Si concentrations in potato leaves. While some other greenhouse experiments corroborated the beneficial effects of Si (soil and foliar) fertilization on potato growth [26,27,28,29,30], Vulavala et al. [31] found no significantly changed silica accumulation in potato roots or leaves after Si fertilization, although they observed an upregulated expression of a gene (called *StLsi1*), encoding a corresponding Si-influx protein in these plant organs.

Notably, field experiments on Si fertilization of potatoes were performed only in a few studies, which were mostly limited to the foliar application of Si in the temperate zone [32,33,34]. One of the rare studies that analyzed soil Si fertilization of potatoes under field conditions was conducted in the tropics by Nyawade et al. [35], who reported synergistic effects of soil Si fertilization and potato–legume intercropping in Kenya. Moreover, the previous studies mainly focused on the effects of Si (soil/foliar) fertilization on potato production using specific plant growth indicators like leaf numbers/areas, protein/saccharide concentrations in leaves, or tuber dry weights. However, the accumulation of silica in specific plant organs on a cellular level has not been the focus of research until now, although Si concentrations in potato plant shoots/organs or tubers were also reported in some of the previous studies (e.g., [25,27,28]).

In our study, we used a scanning electron microscope (SEM) equipped with an energy-dispersive X-ray spectroscopy (EDX) instrument to analyze silica accumulation in potato plants (leaves, roots, tuber flesh, and tuber skin) on a cellular level. The potato plants were taken from a field experiment with control (no Si addition) and Si (addition of artificial silica to the soil) plots and Si concentrations in the microscopically examined plant materials were also determined spectroscopically in corresponding plant extracts. To gain further insights into the effects of Si on potato growth, we additionally used potato yield data from an ongoing long-term field experiment (LTFE) with plots where plant-available Si in soils has been increased via crop straw recycling. The combination of microscopical and spectroscopical techniques, as well as the combined analysis of results from two different field experiments in our study, will help us to evaluate the effects of Si supply on potato cultivation in detail. The corresponding results will not only be interesting for agricultural scientists, but also for potato growers worldwide.

## 2. Materials and Methods

### 2.1. Study Sites and Sampling

The two field experiments, “V434” (silica amendment experiment) and “V140” (LTFE), are located in the experimental area of the Leibniz Centre for Agricultural Landscape Research (ZALF). The experimental sites are managed according to “Good Agricultural Practice”. The climate is characterized by a mean annual precipitation of 535 mm and a mean annual temperature of 9.3 °C based on the reference period of 1991–2020 (measured by a weather station of the German Meteorological Service installed on the ZALF area).

The silica amendment experiment was established in 2020 [36]. This experimental site consists of 12 plots measuring 3 m × 4 m each. Six of these plots serve as control and received no Si addition. The soil at six other plots was amended with artificial amorphous silica (ASi; Aerosil 300, Evonik Industries, Essen, Germany) in the upper 25 cm (Ap horizon) in different amounts. While the soil at three of the Si plots was mixed with 1.8 kg ASi m^−2^, resulting in a mass percentage of 0.5%, the soil at the three other Si plots was mixed with 3.6 kg ASi m^−2^, resulting in a mass percentage of 1.0%. A block design was used for practical purposes, with buffer strips of equal size between the Si-treated and control plots to prevent cross-treatment interferences (Figure 1). The ASi was carefully mixed into the soil by hand in the first step and, subsequently, a cultivator was used to homogenously distribute the ASi in the topsoil. To ensure comparable soil conditions, the soil at the control plots was also treated with the cultivator, but without mixing in ASi. Finally, an overhead sprinkler system was used to irrigate all plots with 60 mm m^−2^ of water, ensuring uniform soil moisture across the entire field. In the first two years, wheat (*Triticum aestivum*) was cultivated on the different plots of the silica amendment experiment [36,37]. In June 2020, soil samples were taken and prepared (i.e., air-dried and passed through a 2 mm sieve) for further analyses, i.e., the extraction of plant-available Si (see Section 2.2).

Seed potatoes (cultivar “Talent”) were planted on 22 April 2022. In May 2022, a mineral fertilizer (“Piasan 25/6”; 120 kg nitrogen ha^−1^ and 29 kg sulfur ha^−1^) was applied. Pesticides were applied in May (herbicide) and June (insecticide, fungicide) 2022. Potato plant samples (shoots, roots, and tubers) were carefully taken on two dates in 2022, i.e., on the 30 June and the 28 July. On the first date, several specimens of the ten-lined potato beetle (*Leptinotarsa decemlineata* Say, 1824) were already observed. Despite a three-time application of insecticides, the beetle infestation increased and, on the second date, the shoots of the potato plants were heavily damaged. Plant samples were thoroughly washed, dried, and, finally, used for Si analyses (see Section 2.2).

The LTFE (52°31′01″ N, 14°07′19″ E) at ZALF was established in a full randomized block design in 1963 to analyze the effects of different mineral and organic fertilizers on yields and soil fertility [21,38]. The sand-dominated soil is classified as Albic Luvisol (Arenic, Neocambic [39]), with two argic horizons in depths of 80–120 cm (Bt1) and 120–160 cm (Bt2). The experimental setup includes different treatments, i.e., (i) NPK fertilization in steps of 5 rates related to N (plots NPK 1–5), (ii) organic fertilization (manure or straw plots), and (iii) control plots, with 8 field repetitions per treatment (168 single plots in total). Soil samples have been regularly taken by the staff of the Experimental Station of ZALF and analyzed (e.g., pH, phosphate concentrations) in the Central Laboratory of ZALF. Plant biomass (yield) per plot has been determined every year. The effect of crop straw recycling on anthropogenic desilication was analyzed in detail in a previous study using selected plots of the LTFE (i.e., control, NPK 1, NPK 1 + Straw, NPK 3, NPK 3 + Straw, NPK 5, and NPK 5 + Straw plots; see Figure 2) [21]. For our study, we used published (plant-available Si concentrations in soils of the different treatments; samples from 1976, 1998, and 2018 [21]) and unpublished (potato yields for the corresponding treatments; stated for all years, in which potatoes were grown at the LTFE, i.e., for 1965, 1967, 1973, 1983, 1987, 1991, 1999, 2007, and 2015) data to examine the long-term effects of Si supply on potato yield performance. Monthly temperature and precipitation data (1965–2015) were used to analyze the potential effects of climate change on potato yields.

### 2.2. Soil and Plant Analyses

Plant-available Si in soil samples of the silica amendment experiment was extracted following the procedures described by Haysom and Chapman [40] and de Lima Rodrigues et al. [41]. In short, two-gram samples of soil were placed in 50 mL plastic centrifuge tubes, mixed with 20 mL of a 0.01 M calcium chloride (CaCl_2_) solution, and agitated continuously on a swivel roller mixer for 16 h. Finally, the extracted solutions were centrifuged at 4000 revolutions per minute (equal to a relative centrifugal force (RCF) of ≈1700) for 30 min and filtrated using 0.45 μm polyamide membrane filters (Whatman NL 17). Si concentrations in the CaCl_2_ extracts were measured via inductively coupled plasma optical emission spectrometry (ICP-OES; iCAP 6300 Duo, Thermo Scientific, Waltham, MA, USA) in the ZALF Central Laboratory.

Si was extracted from plant materials of the silica amendment experiment following the procedure described by Puppe et al. [42]. In short, 30 mg of plant samples were weighed into 50 mL centrifuge tubes, and a 30 mL aliquot of the Tiron solution (pH 10.5) was added. The tubes were then heated at 80 °C in a water bath for 1 h. The samples were gently shaken by hand twice, one time directly before heating and one time after 30 min in the heated water bath. Finally, the extracted solutions were centrifuged at 1700 RCF for 30 min and filtrated (0.45 μm polyamide membrane filters, Whatman NL 17). Si concentrations in the Tiron extracts were measured via microwave plasma atomic emission spectroscopy (MP-AES; 4210 MP-AES instrument, Agilent Technologies Inc., Waldbronn, Germany) following the procedure described by Puppe et al. [43].

All analyses were performed in two lab replicates and three single ICP-OES or MP-AES measurements per replicate resulting in six (n = 6) measured data per sample. Blank sample Si concentrations were subtracted from sample Si concentrations and Si contents in plant samples were calculated considering the weighed portion (2 g or 30 mg), the extractant volume (20 mL or 30 mL), and the degree of dilution (1:10). To avoid any potential Si contamination only plastic equipment was used during the entire laboratory work.

Si analyses on a cellular level were performed using a SEM (ZEISS EVO MA10) equipped with an element detector for EDX (Bruker QUANTAX EDS). We used plant materials from both sampling dates (i.e., 30 June and 28 July 2022) to analyze potential plant growth-related changes in Si accumulation. However, for the first sampling date, only leaf samples were analyzed to check if Si is translocated from the roots to the shoots. For the second date, potato leaves, tubers (tuber skin and tuber flesh), and roots were analyzed. At each sample, several regions of interest were analyzed via SEM-EDX (EDX spectra and compositional maps for Si) to obtain a reliable data set. All SEM-EDX scans were performed using samples sputter-coated with gold (coating thickness approx. 5 nm) and the relative abundances of detected elements were displayed as normalized mass percent.

### 2.3. Statistical Analyses

Linear and monotonic relationships in the data set were analyzed via Pearson’s (*r*) and Spearman’s rank (*r_s_*) correlations (α level of 0.05), respectively. Differences between means were tested with the Mann–Whitney *U* test or the Kruskal–Wallis analysis of variance (ANOVA) followed by pairwise multiple comparisons (Dunn’s post hoc test). All statistical analyses were performed using the software package SPSS Statistics (version 22.0.0.0, IBM Corp., Armonk, NY, USA).

## 3. Results

### 3.1. Silica Accumulation in Potato Plants—Results from the Silica Amendment Experiment

Concentrations of plant-available Si in soils of Si plots (mean for 0.5% Asi, 11.0 mg kg^−1^; mean for 1.0% Asi, 13.5 mg kg^−1^) were significantly higher than in soils of control plots (mean: 4.6 mg kg^−1^). However, differences in concentrations of plant-available Si in soils of 0.5% ASi and 1.0% ASi plots were not statistically significant (Figure 3). The differences in Si availability were directly reflected in the potato leaves collected at the first sampling date: In leaves from plants grown at control plots, we found a relative Si abundance of about 0.2%, while relative Si abundances in leaves of 0.5% ASi and 1.0% ASi plots were about 0.5% and 0.7%, respectively (SEM-EDX measurements, see Figure 4). In leaves of the 1.0% ASi plots collected at the second sampling date, relative Si abundances were even higher (1.1%), indicating a plant growth-related Si accumulation. Relative Si abundances in tuber skin (0.16%), tuber flesh (0.04%), and roots (0.14%) were relatively low in the plants collected at the second sampling date. Selected micrographs and EDX spectra, as well as an exemplary compositional map of our SEM-EDX analyses, can be found in Figure 5. In general, we found only slight silicification on a cellular level in all cross-sections of all analyzed potato plant samples, which was directly related to Si availability (cf. Figure 4). However, no recognizable phytoliths were observed at all. The results of our SEM-EDX analyses were generally corroborated by our Si extraction results: Leaves and roots showed a plant growth-related Si accumulation, i.e., Si contents at sampling date 1 were lower compared to the ones at sampling date 2 with only one exception (i.e., for root samples from 1.0% ASi plots, see Table 1). Moreover, Si accumulation again reflected the Si availability in the soil: Lowest Si contents were detected in plant samples collected at control plots, while Si contents in plant materials collected at Si plots were higher (control < 0.5% ASi < 1.0% ASi). However, as Si contents were relatively inhomogeneous in the analyzed samples (reflected in relatively high standard deviations), we found neither statistical significance for the growth-related differences (Mann–Whitney U test, *p* > 0.05 for control_sampling date 1_ vs. control_sampling date 2_, 0.5% ASi_sampling date 1_ vs. 0.5% ASi_sampling date 2_, and 1.0% ASi_sampling date 1_ vs. 1.0% ASi_sampling date 2_) nor between the different treatments at the two sampling dates (Kruskal–Wallis ANOVA, *p* > 0.05 for control_sampling date 1_ vs. 0.5% ASi_sampling date 1_ vs. 1.0% ASi_sampling date 1_ and control_sampling date 2_ vs. 0.5% ASi_sampling date 2_ vs. 1.0% ASi_sampling date 2_). For tuber skin and tuber flesh samples Si contents were all below the detection limit.

### 3.2. Si Effects on Potato Yields—Results from the Long-Term Field Experiment

In general, potato yields showed a decreasing trend within the analyzed 50-year period at low, medium (i.e., common), and high fertilization plots. At low fertilization plots, yields at NPK + Straw plots were statistically significantly higher than yields at control plots in 6 out of 9 years (Figure 6A). At medium (i.e., common) fertilization plots, yields at NPK + Straw plots were statistically significantly higher than yields at control plots in 4 out of 9 years (Figure 6B). At high fertilization plots, yields at NPK + Straw plots were statistically significantly higher than yields at control plots in 8 out of 9 years (Figure 6C). Moreover, yields at NPK + Straw plots were slightly higher than at NPK plots in 6, 8, and 5 out of 9 years at low, medium (i.e., common), and high fertilization plots, respectively. However, these differences were not statistically significant.

Plant-available Si in soils increased at all plots with experiment duration, especially at NPK + Straw plots. Compared to a mean of 6.3 mg plant-available Si kg^−1^ soil in the year 1976 (range: 5.1–7.6 mg Si kg^−1^, see Puppe et al. [21]), means of plant-available Si in soils increased to 7.2 mg Si kg^−1^ (range: 5.9–8.4 mg Si kg^−1^) and 9.2 mg Si kg^−1^ (range: 8.3–9.9 mg Si kg^−1^) in the years 1998 and 2018, respectively. However, this time-related increase in plant-available Si in soils was statistically significant only at NPK 3 + Straw and NPK 5 + Straw plots (Figure 7). Plant-available Si in soils (data from the years 1976, 1998, and 2018; see Figure 7) and potato yields (data from the years 1973, 1999, and 2015; see Figure 6B,C) at these plots showed low to moderate negative correlations, which were not statistically significant (for NPK 3 + Straw: *r* = −0.436, *p* = 0.713 and *r_s_* = −0.500, *p* = 0.667; for NPK 5 + Straw: *r* = −0.371, *p* = 0.758 and *r_s_* = −0.500, *p* = 0.667).

In the region, where our study sites are located, mean annual temperatures increased from 8.0 °C in the year 1965 to 10.8 °C in the year 2015 (1967: 9.9 °C, 1973: 9.0 °C, 1983: 10.0 °C, 1987: 7.7 °C, 1991: 9.2 °C, 1999: 10.2 °C, 2007: 10.7 °C). This increase was also reflected in elevated temperatures in the potato growing season (April–September) in Brandenburg, Germany (Figure 8). Mean growing season temperatures (1965: 14.1 °C, 1967: 15.4 °C, 1973: 15.0 °C, 1983: 16.2 °C, 1987: 14.2 °C, 1991: 15.0 °C, 1999: 16.5 °C, 2007: 16.5 °C, 2015: 16.2 °C) were moderately highly (*r* = 0.60–0.79) negatively correlated with corresponding potato yields for most plots (i.e., control: *r* = −0.680, *p* = 0.044; NPK 1 + Straw: *r* = −0.728, *p* = 0.026; NPK 5: *r* = −0.687, *p* = 0.041). For NPK 1, NPK 3, and NPK 3 + Straw plots, we found moderate (*r* = 0.40–0.59) to moderately high negative correlations at an α level of 0.10 (i.e., NPK 1: *r* = −0.617, *p* = 0.077; NPK 3: *r* = −0.608, *p* = 0.082; NPK 3 + Straw: *r* = −0.585, *p* = 0.098). Yields at NPK 5 + Straw plots were not statistically significantly correlated with the mean growing season temperatures (*r* = −0.360, *p* = 0.342).

## 4. Discussion

Due to the fact that plant species from the family Solanaceae are considered non-Si-accumulating in general [24], our results showing only quite low Si accumulation in potato plant samples are not surprising at all. Compared to strong Si-accumulating crops like wheat (*Triticum aestivum*, mean shoot Si concentration of about 2.5% in the dry mass) or rice (*Oryza sativa*, mean shoot Si concentration of about 4.2% in the dry mass) [45], we found Si accumulation in potato leaves (max. Si concentration of about 0.08% in the dry mass) and roots (max. Si concentration of about 0.3% in the dry mass) to be about 30–50 or 8–14 times lower, respectively. For tuber skin and tuber flesh samples, Si contents were even below the detection limit of the used MP-AES (i.e., 7.9 μg L^−1^ for Si) [43]. In general, the Si contents we found are within the range of Si contents stated in previous studies [25,26,27,28,31,46]. However, this range is quite big, spanning from 0.2 to 2000 mg Si kg^−1^ dry mass in potato tubers, representing a difference of four orders of magnitude, for example (Table 2).

As the Si contents in our study were at the bottom of the reported Si content range, most previous studies showed considerably higher Si contents in potato plant materials. Vulavala et al. [31], for example, found considerably higher Si contents in potato leaves (about 0.15–0.24% Si in the dry mass), roots (about 1.6–4.4% Si in the dry mass, but results were most likely biased by contaminations with the Si-rich growth medium perlite), and tuber skin (peel, 0.1–0.4% Si in the dry mass) samples collected from control and Si treatments in a pot experiment. In another pot experiment, Crusciol et al. [25] found Si contents in potato leaves to be about 0.4% in the dry mass, which is five times higher than in our study. Soratto et al. [27] reported even higher Si contents in potato plants (for roots up to 1.2% and for shoots up to 1%), which are slightly higher than the mean Si content of maize (*Zea mays*) shoots (0.8%) [45].

In two of the previous studies [26,27] the identical potato cultivar (“Agata”) was examined showing comparable Si contents. This indicates that the reported Si contents in potato plant materials seem to be directly related to the potato cultivar (Table 2). In this context, the differences in reported potato Si contents might be mainly related to the ability of different potato cultivars to take up and transport silicic acid. This ability, in turn, is directly related to the presence of transport/channel proteins that allow silicic acid transportation in the plant [47,48,49]. In general, several influx (called low silicon “Lsi” 1 and Lsi 6) and efflux (Lsi2 and Lsi3) proteins for the transport of silicic acid have been described for rice (Poaceae), but also some other plants like horsetail (Equisetaceae), strawberry (Rosaceae), tomato (Solanaceae), or pumpkin (Cucurbitaceae) [48]. While Lsi1 and Lsi6 represent specific aquaporins that belong to the Nodulin-26-like Intrinsic Proteins (NIPs), Lsi2 and Lsi3 are members of the anion transporter superfamily. The localization of influx and efflux proteins in planta and the expression of corresponding protein-encoding *Lsi* genes control silicic acid transport [50].

Regarding potato plants (*Solanum tuberosum*), Vulavala et al. [31] found *Lsi1* genes (i.e., *StLsi1*) to be expressed in roots and leaves, whereby gene expression was more pronounced in Si compared to control treatments. Expression of the *StLsi2* gene was observed in all potato materials (tuber flesh and skin, stolon, root, stem, and leaf samples) analyzed by these authors, whereby no differences between gene expression in control and Si treatments were observed. However, although Vulavala et al. [31] observed an upregulated expression of *StLsi1* genes in potato roots and leaves (cultivar “Winston”), they found no significantly changed silica accumulation in these plant organs after Si fertilization. Based on their results, Vulavala et al. [31] concluded that the space of 109 amino acids between the asparagine–proline–alanine (NPA) motifs (aquaporins are characterized by two highly conserved hydrophobic NPA motifs, which form a pore or channel for water and/or small molecules like glycerol, urea, or silicic acid) in *StLsi1* explains the low Si accumulation in their potato samples.

This is underpinned by a study by Deshmukh et al. [51], who showed that the ability of plants to take up silicic acid is related to a precise distance of 108 amino acids between the NPA motifs. In total, they analyzed the genomes of 25 plant species including 2 lower plant species (*Physcomitrella patens* and *Selaginella moellendorffii*), 1 gymnosperm species (*Picea abies*), 7 monocot species (e.g., *Oryza sativa*, *Sorghum bicolor*, and *Zea mays*), and 15 dicot species (e.g., *Arabidopsis thaliana*, *Glycine max*, *Solanum tuberosum*, and *Solanum lycopersicum*). Their results showed that Si-accumulating plants had a precise distance of 108 amino acids between the NPA motifs, while plants with 107 or 109 amino acids between the NPA motifs were not able to take up silicic acid in higher amounts. For the wild tomato species *Solanum pimpinellifolium* (Solanaceae), these authors found 109 amino acids between the NPA motifs as well. From their findings, Deshmukh et al. [51] hypothesized that this distance of 109 amino acids most likely originates not from a domestication-related genome alteration in cultivated Solanaceae species, but has its origin in the genome of the wild ancestors. However, this hypothesis is derived from the analysis of only one Solanaceae species (wild tomato), and further research is necessary to draw general conclusions regarding this aspect.

Regarding potatoes, it is assumed that the more than 4000 cultivars globally known originate from a relatively small sample of South American clones only, but with a relatively large amount of genetic diversity [52]. Thus, it cannot be ruled out that there might be differences in NPA motif amino acid distances between different potato cultivars controlling their ability to take up silicic acid. What we need now are detailed genome analyses (NPA motifs) of the various potato cultivars grown worldwide to clarify this aspect. Moreover, Thorne et al. [53] recently found that different, widely cultivated rice cultivars grown under hydroponic conditions showed different, cultivar-specific shoot and root Si concentrations, which were dependent on the levels of sodium chloride (salinity stress) and Si (plant Si availability). As plant Si availability is another crucial factor for Si uptake by plants [18], the relationship between the ability of specific potato cultivars to take up silicic acid and the concentrations of plant-available Si in agricultural soils has to be considered in future studies as well. Combined potato cultivar genome and soil Si availability studies will allow us to better understand the cultivar-specific differences in the uptake of silicic acid and to derive corresponding practice-oriented recommendations for potato growers worldwide.

The potato yields at our LTFE showed a decreasing trend within the analyzed 50-year period at low, medium, and high fertilization plots. We attribute this yield decrease to climate change to a certain degree, because the yield of many potato genotypes is quite sensitive to elevated temperatures, as potatoes originate from the Andes in South America, i.e., from a region with relatively cool temperatures. In fact, temperatures above 17 °C lead to a diminishment of potato tuberization, and thus global warming has been predicted to lead to decreased potato yields on a global scale in general [54,55]. In the region, where our study sites are located, mean annual temperatures increased from 8.0 °C in the year 1965 to 10.8 °C in the year 2015. This increase was also reflected in elevated temperatures in the potato growing season (April–September) in Brandenburg, Germany, which were negatively correlated to potato yields. However, it has to be stated here that we do not know to which extent other climate-related factors (e.g., drought, pest infestation, or heavy precipitation) and/or changes in soil properties (e.g., soil moisture, soil organic matter, or soil pH) (cf. [56,57,58,59]) affected potato yields at our experimental fields. The evaluation of such interactions was outside the scope of our study, which aimed at the analysis of silica accumulation in potato plants and the relationship between plant-available Si in agricultural soils and corresponding potato yield performance in the long term.

We found no relationship between the concentration of silicic acid (plant-available Si) in soils and corresponding potato yields in our study at all. Based on our (long-term) results and because Si contents of potato plant materials from control and Si treatments often show no statistically significant differences (see Table 2), we assume that silica accumulation in potato plants has no effect on potato yield performance. Consequently, we ascribe the reported (beneficial) effects of Si fertilization on potato growth and yield performance [26,27,28,29,30,32,34] mainly to antifungal/osmotic effects of foliar-applied Si fertilizers [16] and to changes in physicochemical soil properties (e.g., enhanced phosphorus availability and water-holding capacity) caused by soil-applied Si fertilizers [36,60]. In fact, potato plants can suffer from numerous diseases, which are caused by fungi (e.g., *Alternaria solani* (early blight), *Rhizoctonia solani* (black scurf), *Synchytrium endobioticum* (black scab), or *Fusarium* spec. (colored rots)) or fungus-like microorganisms (e.g., *Phytophthora infestans* (late blight)) in most cases [61,62]. Moreover, phosphorus and water availability in agricultural soils represent the main controls for potato growth and yield, because potatoes are characterized by a relatively high phosphorus requirement and susceptibility to even mild water stress [63,64,65,66]. However, as research on the effects of Si fertilization on potato performance is still limited to a few potato cultivars (cf. Table 2), we are calling for more studies dealing with the aspects discussed above.

## 5. Conclusions

Based on our results, we assume that the beneficial effects of Si fertilization on potato growth and yield performance reported in previous studies are related to the effects of the used Si fertilizers, rather than to silica accumulation in potato plants. In this context, antifungal/osmotic effects of foliar-applied Si fertilizers and changes in physicochemical soil properties (e.g., enhanced phosphorus availability and water-holding capacity) caused by soil-applied Si fertilizers seem to be the strongest candidates to explain the phenomena observed. To derive practice-oriented recommendations for potato growers worldwide, future research should aim at elucidating the complex relationships between the cultivated potato cultivar, the used Si fertilizer, and the prevalent soil properties as well as climate conditions. In this context, the following questions might be of particular interest:
(i)How big is the range of Si contents in potato plants considering the numerous cultivars worldwide? Recently, published data show that Si contents in potato tubers represent a difference of four orders of magnitude, for example (Table 2).(ii)Which foliar Si fertilizer formula, at which dose, is most effective against which disease caused by fungi or fungus-like microorganisms?(iii)How do different soil Si fertilizers (e.g., slags, fused magnesium phosphate, wollastonite, or biochar) affect soil properties in different soils under different climate conditions?

## Figures and Tables

**Figure 1 biology-13-00828-f001:**
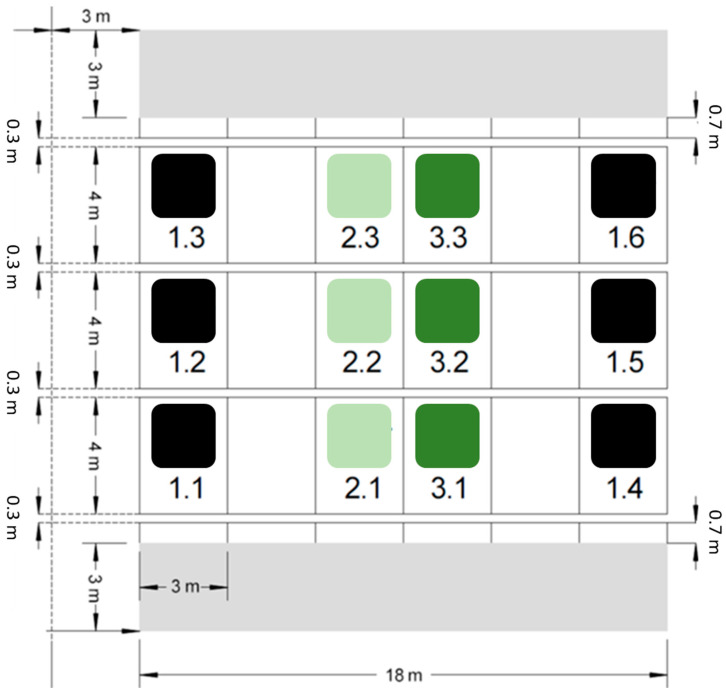
Overview of the plots at the silica amendment experiment. While six plots serve as control (i.e., plot numbers 1.1–1.6, marked by black squares), six plots represent Si treatments with 0.5% (i.e., plot numbers 2.1–2.3, marked by light green squares) or 1.0% (i.e., plot numbers 3.1–3.3, marked by dark green squares) amorphous silica mass percentage.

**Figure 2 biology-13-00828-f002:**
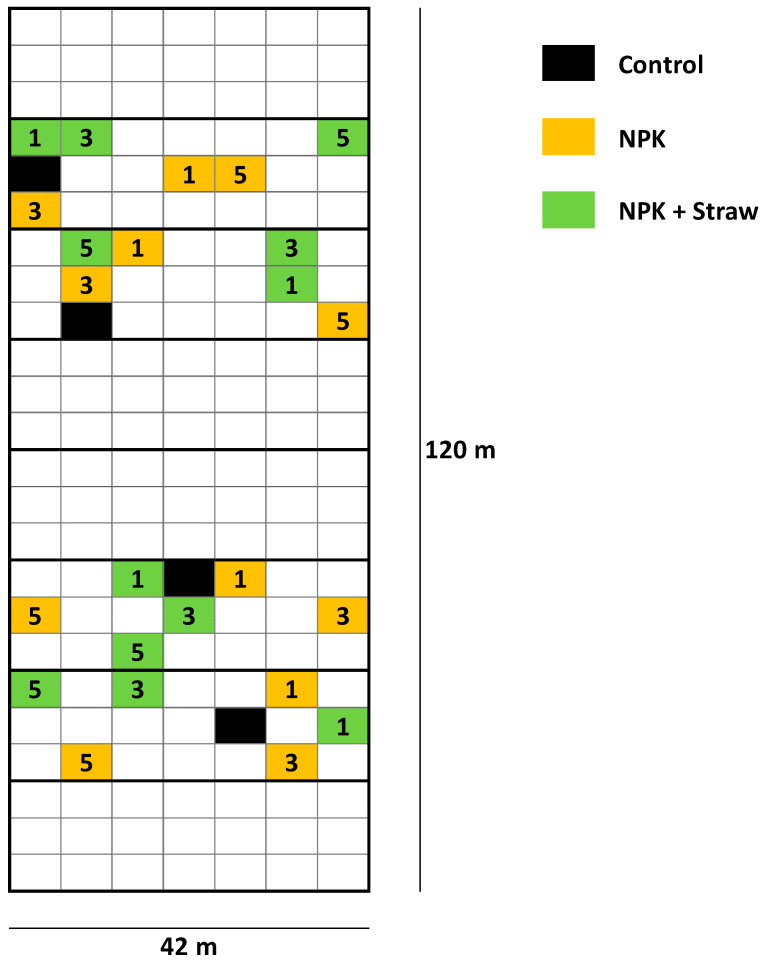
Overview of the plots at the LTFE (modified from Puppe et al. [21]). The plots used in the study by Puppe et al. [21] are highlighted in color (see legend). 1 = low fertilization rate (NPK 1, ~30 kg N ha^−1^ y^−1^), 3 = medium (i.e., common) fertilization rate (NPK 3, ~98 kg N ha^−1^ y^−1^), and 5 = high fertilization rate (NPK 5, ~166 kg N ha^−1^ y^−1^). At plots with crop straw recycling (NPK + Straw), NPK fertilization has been supplemented by incorporation of 4.0 t (dry mass) straw ha^−1^ every second year using chopped straw of the recently harvested cereal crop. At the control plots, neither NPK fertilization, nor crop straw recycling has been performed.

**Figure 3 biology-13-00828-f003:**
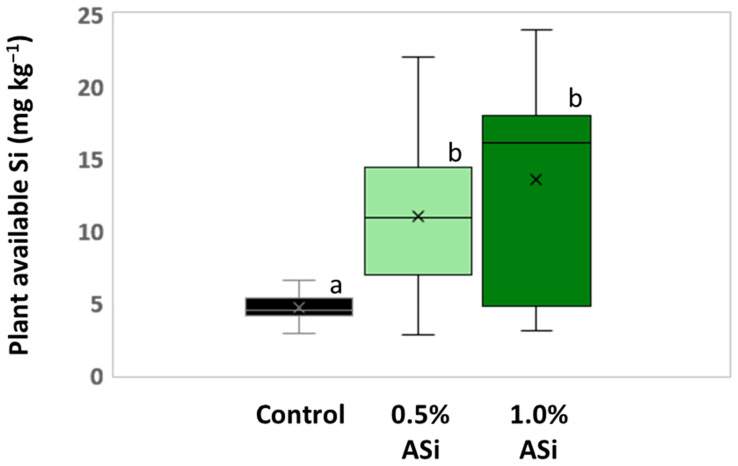
Plant-available Si in soils of control and Si plots of the silica amendment experiment. Means are marked by “x” in the boxplots each. Different letters indicate statistically significant differences (Kruskal–Wallis ANOVA, *p* < 0.05) between the plots.

**Figure 4 biology-13-00828-f004:**
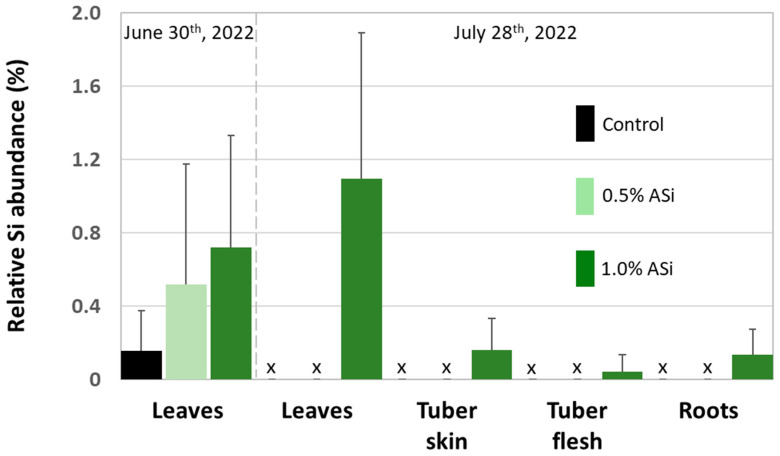
Relative Si abundance (SEM-EDX) in leaves, tubers (i.e., tuber skin and tuber flesh), and roots of potato plants taken at control and Si plots of the silica amendment experiment. Black and green bars represent means of normalized mass percent, error bars represent corresponding standard deviations. x = no data available.

**Figure 5 biology-13-00828-f005:**
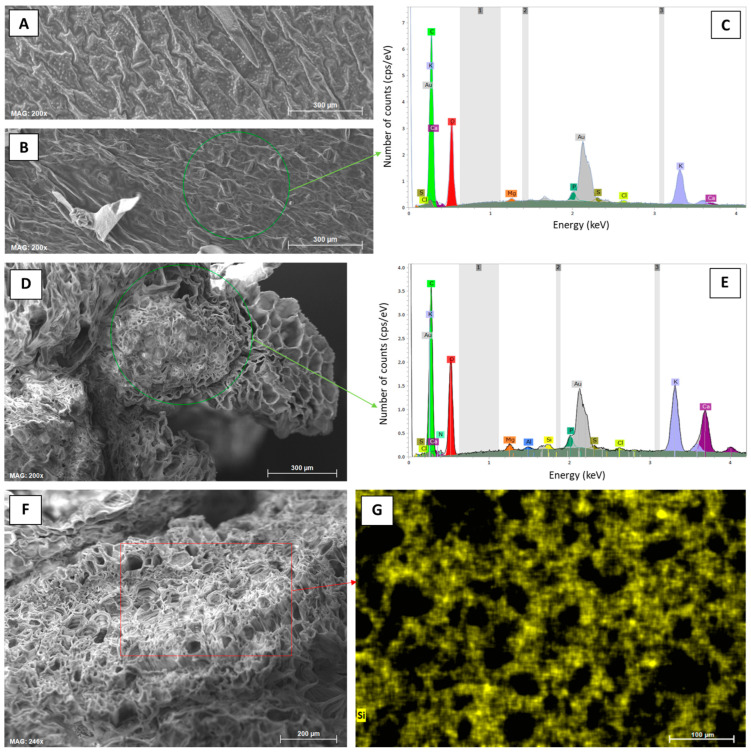
Elemental analyses (SEM-EDX) of leaf samples from potato plants taken at the first sampling date (30 June 2022) at control and Si plots of the silica amendment experiment. (**A**) Micrograph of the leaf top epidermis (control), (**B**) micrograph of the leaf undersurface epidermis (control), (**C**) corresponding exemplary EDX spectra derived from SEM-EDX measurements performed in a specific region of interest in (**B**) (green circle), (**D**) micrograph of a leafstalk cross-section (1.0% ASi), (**E**) corresponding exemplary EDX spectra derived from SEM-EDX measurements performed in a specific region of interest in (**D**) (green circle), (**F**) micrograph of a leafstalk cross-section (1.0% ASi), and (**G**) corresponding compositional map for Si in a specific region of interest in (**F**) (red rectangle).

**Figure 6 biology-13-00828-f006:**
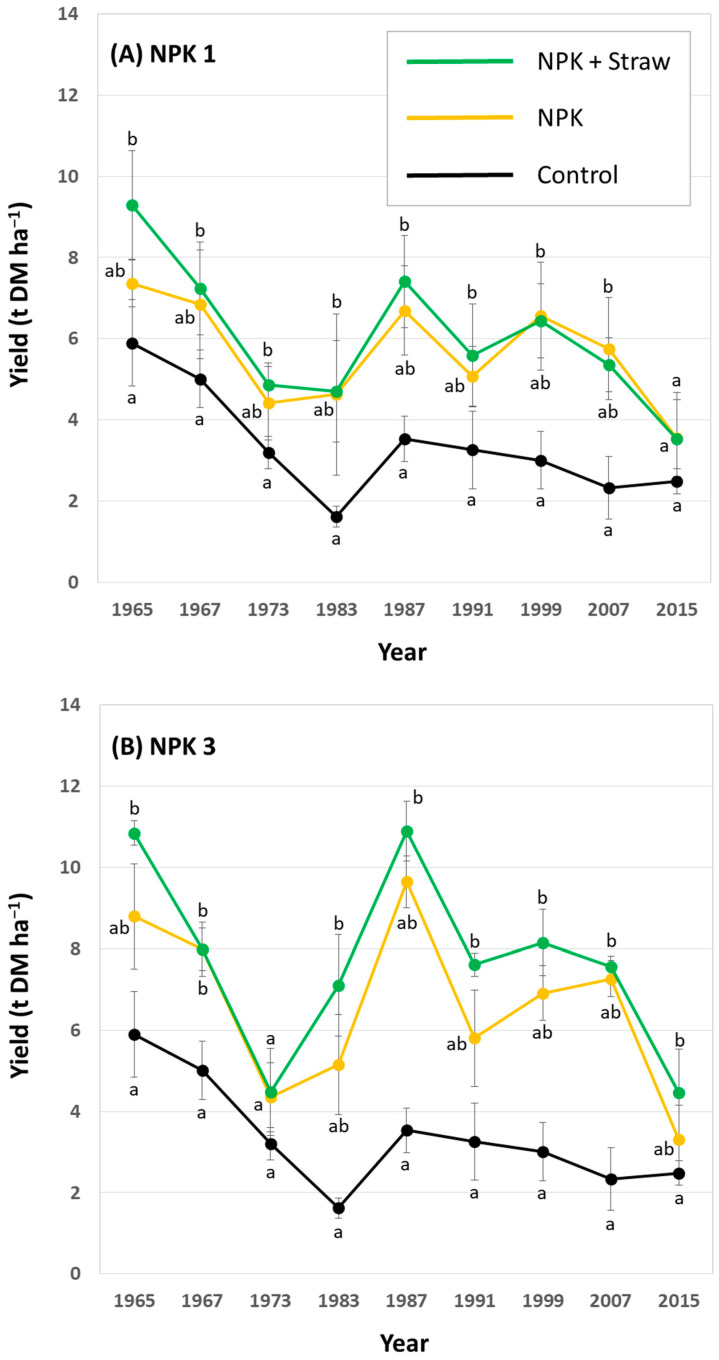
Potato yields for low (**A**), medium (i.e., common) (**B**), and high (**C**) fertilization plots (NPK 1, NPK 3, and NPK 5, respectively) at the long-term field experiment. Yields are stated for all years in which potatoes were grown during the ongoing long-term field experiment. Different letters indicate statistically significant differences (Kruskal–Wallis ANOVA, *p* < 0.05) between control, NPK, and NPK + Straw plots in a specific year.

**Figure 7 biology-13-00828-f007:**
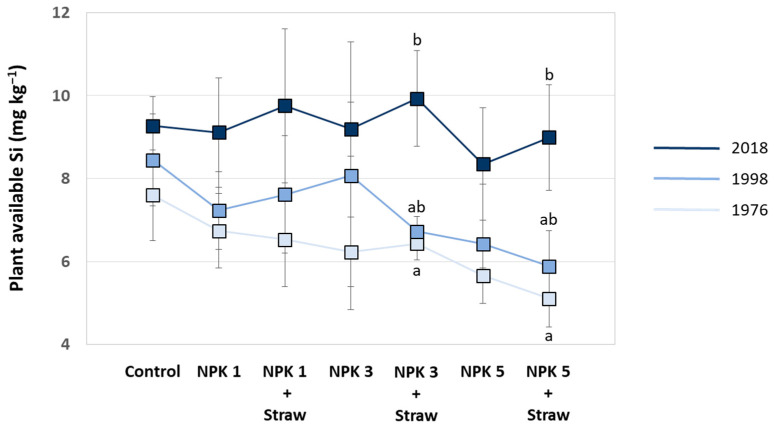
Plant-available Si in soils of the different plots at the long-term field experiment for the years 1976, 1998, and 2018 (data taken from Puppe et al. [21]). Different letters indicate statistically significant differences (Kruskal–Wallis ANOVA, *p* < 0.05) between the three years for specific plots. If no statistical significances were found for a specific plot, no letters were stated.

**Figure 8 biology-13-00828-f008:**
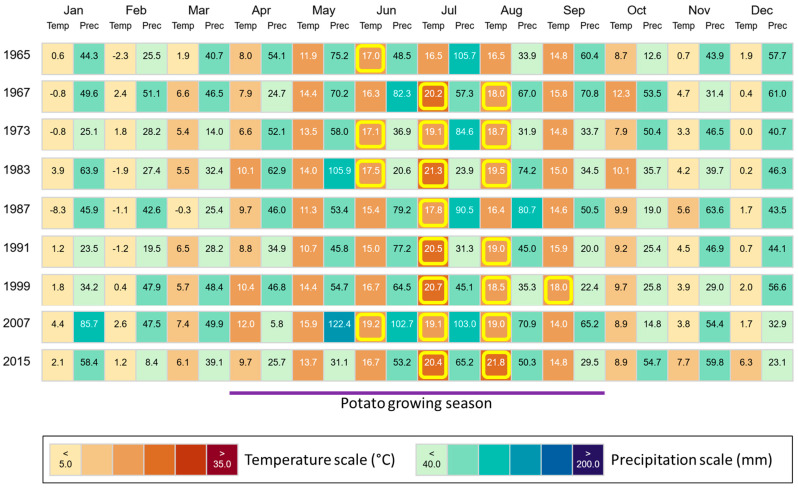
Monthly climate data (temperature and precipitation) for the region, where our study sites are located. Climate data are stated for all years in which potatoes were grown during the ongoing long-term field experiment at ZALF. Temperatures ≥ 17 °C (diminishment of potato tuberization) in the potato growing season (April–September) in Brandenburg, Germany, are highlighted in yellow. Figure created using “ClimateCharts.net” [44], modified.

**Table 1 biology-13-00828-t001:** Si contents (Tiron extraction) in leaves, tubers (i.e., tuber skin and tuber flesh), and roots of potato plants taken at control and Si plots of the silica amendment experiment. SD = standard deviation.

		Si Content (mg kg^−1^)			
		30 June 2022		28 July 2022	
Treatment	Plant Material	Mean	SD	Mean	SD
Control	Leaves	0	--	50	0.2
0.5% ASi	Leaves	0	--	646	--
1.0% ASi	Leaves	12	263	789	--
Control	Tuber skin	0	--	0	--
0.5% ASi	Tuber skin	0	--	0	--
1.0% ASi	Tuber skin	0	--	0	--
Control	Tuber flesh	0	--	0	--
0.5% ASi	Tuber flesh	0	--	0	--
1.0% ASi	Tuber flesh	0	--	0	--
Control	Roots	316	405	860	929
0.5% ASi	Roots	936	762	1669	2361
1.0% ASi	Roots	3198	2081	2401	3326

--: no data available.

**Table 2 biology-13-00828-t002:** Overview of reported Si contents in potato plant materials of various potato cultivars.

			Si Content (mg kg^−1^ DM)		Si Contents of Control and Si Treatments Statistically Significantly Different?	
Year	Potato Cultivar	Plant Material	Control	Si Treatment(s)		Reference
2009	Bintje	Leaves	3700–4100	4200–4700	yes (under drought stress)/no (without stress)	Crusciol et al. [25]
2013	Agata	Leaves	4100	8300–10,000	yes	Pilon et al. [26]
		Stems	6300	7600–10,100	yes (soil Si application)/no (foliar Si application)	
		Roots	3800	4000–5900	yes (soil Si application)/no (foliar Si application)	
		Tubers	2000	2100–2200	no	
2016	Winston	Leaves	1400–2300	1500–2200	no	Vulavala et al. [31]
		Roots ^a^	15,600–41,300	17,300–34,200	no	
		Tuber skin	950–2000	850–3900	no	
2018	Agria	Shoots + roots	26	27–50	ns	Soltani et al. [28]
		Tubers	37	40–46	ns	
2019	Agata	Leaves	8300	8400–8600	no	Soratto et al. [27]
		Roots	11,000	11,600–12,300	no	
		Shoots	8100	8300–9600	yes (high Si fertilization level)/no (low Si fertilization level)	
		Tubers	1200	2100–2300	yes	
2023	Catania	Tubers	0.2	0.3	no	Wadas and Kondraciuk [46]
2024	Talent	Leaves	0–50	0–790	no	This study
		Tuber skin	0	0	no	
		Tuber flesh	0	0	no	
		Roots	320–860	940–3200	no	

^a^ = most likely contaminated with the Si-rich growth medium perlite; DM = dry mass; ns = not specified.

## Data Availability

All relevant data are presented within the paper. Underlying data will be made available by the corresponding author upon reasonable request.

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
