# Peer review of "Silica Accumulation in Potato (Solanum tuberosum L.) Plants and Implications for Potato Yield Performance—Results from Field Experiments in Northeast Germany"

_biology, 2024, doi:10.3390/biology13100828_

Round 1

Reviewer 1 Report

Comments and Suggestions for Authors

The research topic is interesting but the manuscript needs improvement. There are 11 comments marked in the text. However, there are more general comments. There are too many cited items in the introduction. It is not necessary to cite 5 authors for an evident statement. The methodology is written correctly. There are passages in the discussion that are unnecessary because they do not apply to the research being carried out. Conclusions do not relate to the research results obtained. The authors write what should be studied. The last two chapters should be rewritten.

Author Response

Reviewer #1

Thanks a lot for your critical review of our manuscript. Please find our answers to your comments below.

  1. E-mail addresses of the authors are missing.

Response: The email address of the corresponding author is stated in the manuscript.

  1. The keywords are too many. Please reduce.

Response: According to the “Instructions for Authors” of this journal three to ten pertinent keywords need to be added after the abstract. As we state 9 keywords in our manuscript, we do not see any demand for reduction.

  1. protists (i.e., testate amoebae) ?

Response: Yes, protists have been found to play an important role in terrestrial silicon cycles (see Puppe 2020).

  1. without spaces

Response: We adjusted the citation style using the official “MDPI output style” now. Thanks for the hint.

  1. Enter as an item for references.

Response: Modified as suggested. Thanks.

  1. It doesn't take that many items ( 5) for simple information.

Response: Actually, we intended to give a summarizing overview of the current knowledge of the effects of silicon on potatoes in this paragraph. Thus, we would like to keep it as it is.

  1. This passage does not apply to research. It describes other plants. I don't think it is needed.

Response: From our point of view this paragraph is needed to generally explain silicic acid transportation in plants. As not every reader is familiar with this aspect, we would like to keep this paragraph as it is.

  1. This passage does not apply to research. It describes other plants. I don't think it is needed.

Response: Actually, this paragraph is directly related to potato plants, and thus is absolutely needed in the discussion from our point of view.

  1. space

Response: We adjusted the citation style using the official “MDPI output style” now. Thanks for the hint.

  1. The proposals should be amended. They should be a summary of the research results. It is not written in the conclusions what should be studied. This is not on topic.

Response: From our point of view a summary of the article should be given only in the Abstract. The Conclusions should be used to end the main text (as stated in the “Instructions for Authors”). This section should answer the following questions: What can be concluded from the results obtained and what should be done in the future to move the field forward? Thus, we would like to keep this section as it is.

  1. References should be edited according to the editor's requirements. Authors should read the instructions on the journal's website.

Response: Thanks for the hint. We now use the official “MDPI output style” and adjusted the References accordingly.

Thanks again for your comments on our manuscript.

Reviewer 2 Report

Comments and Suggestions for Authors

REVIEW REPORT: BIOLOGY - 3238336

TITLE:

Silica accumulation in potato (Solanum tuberosum L.) plants and implications for potato yield performance – Results from field experiments in NE Germany

AUTHORS:

Daniel Puppe*, Jacqueline Busse, Mathias Stein, Danuta Kaczorek, Christian Buhtz, Jörg Schaller

* Correspondence:  daniel.puppe@zalf.de

Reviewer: ID

Comments and Suggestions for Authors (28/09/2024)

Simple Summary and Abstract:

Very good, sufficient and clear in relation to the research topic and title.

All themes under experimentation were mentioned, explaining the research carried out very well, through the results shown.

The keywords are very well chosen

1. Introduction:

Very good presentation and enough to clarify and understand the rest of the paper. Very good writing, clear and coherent.

Meets the requirements requested by Biology Journal.

The objectives of the article are well referenced.

He explains all the research very well, based on his own experimentation and knowledge. To confirm and strengthen the document, it cites 34 authors.

2. Materials and Methods

Well-chosen sub-chapters, written in great detail and very enlightening, making it easier for the reader to interpret the results on all points.

This chapter meet those requested in the MDPI journals; the authors write only the essential information regarding the results without repetitions.

Very good, it presents a very interesting approach when comparing current data with data from several decades (solidifies information).

3. Results:

The statistical analysis was well chosen.

The interpretation of the results is very well done, very clear and easy to read

The sequence of results is aligned with the Materials and Methods chapter. The subchapter titles are well chosen.

The tables are clear and the figures are very interesting

In lines 158 and 159 there are 3 words highlighted in yellow, why?

4. Discussion

I like that way - how it introduce the data discussion; the authors, that supporting the results, are very well placed.

I also noticed that you repeat, confirming statements made by authors mentioned in the Introduction chapter; very positive for confirming results.

5. Conclusions:

Conclusions are very objective and clear.

They match the results presented in the paper.

Confirms and concludes the results and corresponds to the abstract and Keywords.

Very interesting how the authors end the conclusions, with questions of particular interest, about techniques to be followed in future works.

References:

The work is very strong and rich, as can be seen from the large number of references (65 references and they are very diverse), citing the authors themselves.

However, the authors never reference Biology Journal articles

GENERAL INFORMATION:

The article is very well written, very interesting and is very useful for the international community

Author Response

Reviewer #2

Thanks a lot for your critical review of our manuscript. Please find our answers to your comments below.

REVIEW REPORT: BIOLOGY - 3238336

TITLE: Silica accumulation in potato (Solanum tuberosum L.) plants and implications for potato yield performance Results from field experiments in NE Germany

AUTHORS: Daniel Puppe*, Jacqueline Busse, Mathias Stein, Danuta Kaczorek, Christian Buhtz, Jörg Schaller

* Correspondence: daniel.puppe@zalf.de

Reviewer: ID

Comments and Suggestions for Authors (28/09/2024)

Simple Summary and Abstract: Very good, sufficient and clear in relation to the research topic and title. All themes under experimentation were mentioned, explaining the research carried out very well, through the results shown. The keywords are very well chosen

Response: Thank you very much for your kind words.

  1. Introduction: Very good presentation and enough to clarify and understand the rest of the paper. Very good writing, clear and coherent. Meets the requirements requested by Biology Journal. The objectives of the article are well referenced. He explains all the research very well, based on his own experimentation and knowledge. To confirm and strengthen the document, it cites 34 authors.

Response: Thanks again.

  1. Materials and Methods: Well-chosen sub-chapters, written in great detail and very enlightening, making it easier for the reader to interpret the results on all points. This chapter meet those requested in the MDPI journals; the authors write only the essential information regarding the results without repetitions. Very good, it presents a very interesting approach when comparing current data with data from several decades (solidifies information).

Response: Thanks a lot for your very motivating comments.

  1. Results: The statistical analysis was well chosen. The interpretation of the results is very well done, very clear and easy to read. The sequence of results is aligned with the Materials and Methods chapter. The subchapter titles are well chosen. The tables are clear and the figures are very interesting. In lines 158 and 159 there are 3 words highlighted in yellow, why?

Response: Thank you. The mentioned highlighting was done accidentally. We removed it from our revised manuscript.

  1. Discussion: I like that way - how it introduce the data discussion; the authors, that supporting the results, are very well placed. I also noticed that you repeat, confirming statements made by authors mentioned in the Introduction chapter; very positive for confirming results.

Response: Thanks, we really appreciate your very positive comments on our manuscript.

  1. Conclusions: Conclusions are very objective and clear. They match the results presented in the paper. Confirms and concludes the results and corresponds to the abstract and Keywords. Very interesting how the authors end the conclusions, with questions of particular interest, about techniques to be followed in future works.

Response: Thanks a lot.

References: The work is very strong and rich, as can be seen from the large number of references (65 references and they are very diverse), citing the authors themselves. However, the authors never reference Biology Journal articles

Response: Thank you. You are right, we found no articles on the effects of silicon on potato yield performance in the journal Biology at all. However, there are several articles from other MDPI journals (e.g., Plants) that are cited in our manuscript.

GENERAL INFORMATION: The article is very well written, very interesting and is very useful for the international community

Response: Thanks again. It is a great pleasure to read that you are convinced of the overall-quality of our manuscript.

Reviewer 3 Report

Comments and Suggestions for Authors

The paper examines the features of silicon accumulation in different parts of the potato plant depending on the content of its available forms in the soil.

Lines 20-21: the paper does not disclose the connection between fertilizing potatoes with silicon compounds and climate change. Therefore, this phrase is superfluous.

Lines 30-32: according to the authors, potato yield does not correlate with the content of available forms of silicon in the soil, however, referring to the data of other authors, they indicate the reasons why the yield increases several times. This is a certain contradiction and illogicality of the paper.

Line 453: potato yield is mentioned again in connection with climate change. The authors did not study this issue in their work.

Lines 477-479: the fungicidal effect of silicon compounds is mentioned several times. Is this really true? Again, the authors did not study this issue.

The conclusions of the paper should be based on the results obtained by the authors, so they should be completely rewritten.

I would advise the authors to rely on the differences in the average values ​​of silicon concentrations, even if they are not statistically significant. In this case, we can talk about a "trend".

Author Response

Reviewer #3

Thanks a lot for your critical review of our manuscript. Please find our answers to your comments below.

The paper examines the features of silicon accumulation in different parts of the potato plant depending on the content of its available forms in the soil.

Lines 20-21: the paper does not disclose the connection between fertilizing potatoes with silicon compounds and climate change. Therefore, this phrase is superfluous.

Response: You are right, we did not directly study climate change. However, as we used potato yield data from an ongoing long-term experiment, our results are naturally related to climate data. From our point of view this aspect has been well-inferred from our (and previous) results in the discussion section of our manuscript. Thus, we would like to keep this sentence as it is. However, to meet your concerns we added an explaining sentence in the M&M section.

Lines 30-32: according to the authors, potato yield does not correlate with the content of available forms of silicon in the soil, however, referring to the data of other authors, they indicate the reasons why the yield increases several times. This is a certain contradiction and illogicality of the paper.

Response: From our point of view this is not a contradiction or illogicality, but science. Different studies often show inconsistent results (please see also our Table 2), and of course all these results must be considered to obtain the “big picture”. In our manuscript we clearly state (i) what was reported in previous studies (Introduction), (ii) what we found in our study (Results), and (iii) how all these (contrasting) findings could be explained (Discussion). Thus, we would like to keep this sentence as it is.

Line 453: potato yield is mentioned again in connection with climate change. The authors did not study this issue in their work.

Response: As we used potato yield data from an ongoing long-term experiment, our results are naturally related to climate data. From our point of view this aspect has been well-introduced in our discussion section, where we subsequently discuss our (and previous) results. Moreover, we phrase quite carefully (“…to a certain degree”) to avoid any over-interpretation of our results. Thus, we would like to keep this paragraph as it is. However, to meet your concerns we added an explaining sentence in the M&M section.

Lines 477-479: the fungicidal effect of silicon compounds is mentioned several times. Is this really true? Again, the authors did not study this issue.

Response: Antifungal/osmotic effects of foliar-applied silicon fertilizers have been studied by several authors. A summary can be found in the review article of Puppe and Sommer (2018) (cited in our manuscript in the corresponding paragraph). Thus, we would like to keep this paragraph as it is.

The conclusions of the paper should be based on the results obtained by the authors, so they should be completely rewritten.

Response: From our point of view our Conclusions are absolutely supported by our findings. Of course, this is also always a question of taste. Reviewer #2, for example, wrote: “Conclusions are very objective and clear. They match the results presented in the paper. Confirms and concludes the results and corresponds to the abstract and Keywords. Very interesting how the authors end the conclusions, with questions of particular interest, about techniques to be followed in future works.”

Thus, we would like to keep it as it is.

I would advise the authors to rely on the differences in the average values of silicon concentrations, even if they are not statistically significant. In this case, we can talk about a "trend".

Response: Thanks for the hint. Although we agree that one should not rigidly insist on statistically significant findings only, the results of statistical analyses are an important factor in the interpretation of scientific data. From our point of view, we considered all results (statistically significant or not) in our discussion (please see also Table 2) and always clearly stated how we interpreted the corresponding results. This is also reflected by the corresponding comment of Reviewer #2: “The statistical analysis was well chosen. The interpretation of the results is very well done, very clear and easy to read. The sequence of results is aligned with the Materials and Methods chapter.”

Thanks again for your comments on our manuscript.

Round 2

Reviewer 1 Report

Comments and Suggestions for Authors

In response, the authors explained why they left certain comments uncorrected. This is acceptable. It remains to correct the references. The long dash between page numbers should be there. Titles of journals given what abbreviations.